# Lagrangian Formulation of Free Arbitrary *N*-Extended Massless Higher Spin Supermultiplets in 4D AdS Space

**Ioseph L. Buchbinder** [1,2] **and Timofey V. Snegirev** [1,3,*]

[1]    Department of Theoretical Physics, Tomsk State Pedagogical University, 634061 Tomsk, Russia;
       joseph@tspu.edu.ru
[2]    National Research Tomsk State University, 634050 Tomsk, Russia
[3]    Department of Mathematics and Informatics, National Research Tomsk Polytechnic University,
       634050 Tomsk, Russia
*    Correspondence: snegirev@tspu.edu.ru

**Abstract:** We derived the component Lagrangian for the free *N*-extended on-shell massless higher spin supermultiplets in four-dimensional anti-de Sitter space. The construction was based on the frame-like description of massless integer and half-integer higher spin fields. The massless supermultiplets were formulated for $N \leq 4k$, where $k$ is a maximal integer or half-integer spin in the multiplet. The supertransformations that leave the Lagrangian invariant were found in explicit form and it was shown that their algebra is closed on-shell.

**Keywords:** extended supersymmetry; higher spin symmetry; gauge invariance

## 1. Introduction

The study of various aspects of higher spin fields is currently one of the actively developing areas of modern theoretical and mathematical physics (for review see, e.g., [1–5]). Recently, there has been a surge of interest in constructing supersymmetric higher spin models (in the literature, higher spin models are sometimes called higher spin (super)gravities) and in investigating the properties of such models [6–42]. In this paper, we studied a general problem of Lagrangian construction for arbitrary *N*-extended massless free on-shell supermultiplets in four-dimensional (4D) *AdS* space and derived the Lagrangians describing the dynamics of such supermultiplets.

It is well known that in four dimensions, *N*-extended supermultiplets with maximal spin $k = 1$ are restricted by the condition $N \leq 4$, and that multiplets with maximal spin $k = 2$ are restricted by the condition $N \leq 8$. Supermultiplets with $N > 8$ must contain higher spins of $k > 2$. To be more precise, there is a specific relationship between the parameter *N* and the highest spin $k$ in a supermultiplet, $N \leq 4k$ (see, e.g., [43]). Of course, if one does not restrict the maximal spin in a multiplet by the quantities $k = 1, 2$, then for any *N*, there exists supermultiplets with arbitrary higher spins.

For the case of simple $N = 1$ supersymmetry, the component Lagrangian formulation of on-shell higher spin supermultiplets in Minkowski space has been known for a long time [44,45]; furthermore, the component approach has been generalized and studied in [46–52]. In particular, supertransformations have been found that leave invariant the sum of Lagrangians for free massless fields with spins $k$ and $k + 1/2$. Completely off-shell Lagrangian formulations for such theories have been constructed within the framework of the superfield approach [53,54] (see also the later paper [55] on the same subject). The off-shell formulation of the $N = 1$ higher superspin free Lagrangian theory in 4D AdS space was first developed in [56] at the very beginning of superfield formalism, and its

component form was derived from superfield theory. Quantization of this theory was provided in [57]. The $N = 2$ supersymmetric higher spin models both in the Minkowski and AdS spaces were discussed in [58], while the universal higher spin superfield approach in 4D $N = 1$ AdS superspace was developed in [59].

Recently, on-shell superfield Lagrangian realization was constructed for extended $N = 1$ massless supermultiplets in the framework of light-cone gauge formalism [41]. An extension of this approach for on-shell $N$-extended superfield Lagrangian formulation was given in [42] under the condition $N = 4n$, where $n$ is a natural number. In this paper, we generalized the results of [42] and provide an explicit component Lagrangian construction of arbitrary $N$-extended massless higher spin on-shell supermultiplets in 4D anti-de Sitter space without using the condition accepted in [42].

Our construction was based on the frame-like approach for higher spin fields. The generic scheme of the Lagrangian formulation for free higher spin bosonic and fermionic fields in this approach was developed in [46]. The full higher spin field Lagrangian is the sum of the free Lagrangians for the bosonic and fermionic component fields of an on-shell supermultiplet. The main question that must be solved in such an approach is finding the supersymmetry transformations that leave the full Lagrangian invariant. In principle, the necessary supersymmetry transformations can be obtained on the basis of the construction developed in [47,50]; however, an explicit realization of the supersymmetry transformations has not been derived thus far. Herein, we aimed to fill this gap and to find the explicit supertransformations for $N$-extended massless higher spin supermultiplets that leave the sum of free bosonic and fermionic Lagrangian invariants and show that the algebra of the supertransformations is closed on-shell.

This paper is organized as follows. In Section 2, we describe the basic elements of the frame-like Lagrangian formulation for free massless higher spin fields in 4D AdS space and the 4D multispinor technique. In Section 3, we present minimal massless $N = 1$ supermultiplets [37], which are used as the building blocks to construct $N$-extended supermultiplets. Section 4 is devoted to constructing the arbitrary $N$-extended massless supermultiplets in 4D AdS. For each case, we formulated the field contents and introduced the corresponding field variables. Then, we derived the supertransformations for these supermultiplets and defined the Lagrangian as a sum of the Lagrangians for all of the integer and half-integer spin fields of a given supermultiplet. We proved that such a Lagrangian is invariant under the above transformations. Finally, we showed that the constructed supertransformations form the closed $N$-extended 4D AdS superalgebras.

## 2. Free Higher Spin Fields

In this section, we briefly consider the frame-like Lagrangian formulation of the free massless higher spin fields in 4D AdS space and the corresponding 4D multispinor formalism.

In the frame-like approach, the massless fields with integer spin $k \geq 2$ are described by the dynamical 1-form $f^{\alpha(k-1)\dot\alpha(k-1)}$ and the auxiliary 1-form $\Omega^{\alpha(k)\dot\alpha(k-2)}$, $\Omega^{\alpha(k-2)\dot\alpha(k)}$ (see all notations in the appendix A) [46]. These fields are totally symmetric with respect to the dotted and undotted indices and generalize the tetrad field and Lorentz connection in the frame formulation of gravity. We chose them to be real values, that is, they satisfy the following rules of hermitian conjugation:

$$
\begin{aligned}
(f^{\alpha(k-1)\dot\alpha(k-1)})^\dagger &= f^{\alpha(k-1)\dot\alpha(k-1)}, \\
(\Omega^{\alpha(k)\dot\alpha(k-2)})^\dagger &= \Omega^{\alpha(k-2)\dot\alpha(k)}.
\end{aligned}
$$

The Lagrangian being the differential 4-form in 4D AdS space looks like:

$$
\begin{aligned}
\frac{(-1)^k}{i}\mathcal{L}_k = {} & k\Omega^{\alpha(k-1)\beta\dot\alpha(k-2)}E_\beta{}^\gamma\Omega_{\alpha(k-1)\gamma\dot\alpha(k-2)} - (k-2)\Omega^{\alpha(k)\dot\alpha(k-3)\dot\beta}E_{\dot\beta}{}^{\dot\gamma}\Omega_{\alpha(k)\dot\alpha(k-3)\dot\gamma} \\
& + 2\Omega^{\alpha(k-1)\beta\dot\alpha(k-2)}e_\beta{}^{\dot\beta}Df_{\alpha(k-1)\dot\alpha(k-2)\dot\beta} \\
& + 2k\lambda^2 f^{\alpha(k-2)\beta\dot\alpha(k-1)}E_\beta{}^\gamma f_{\alpha(k-2)\gamma\dot\alpha(k-1)} + h.c.,
\end{aligned}
\tag{1}
$$

where 1-form $e^{\alpha\dot\alpha}$ is the AdS background tetrad, $D$ is the AdS covariant derivative $De^{\alpha\dot\alpha} = 0$, and $E^{\alpha\beta}$ and $E^{\dot\alpha\dot\beta}$ are a double product of $e^{\alpha\dot\alpha}$ (see appendix for details Appendix A). The form of Lagrangian (1) is determined by the invariance under the gauge transformations:

$$
\begin{aligned}
\delta f^{\alpha(k-1)\dot\alpha(k-1)} &= D\xi^{\alpha(k-1)\dot\alpha(k-1)} + e_\beta{}^{\dot\alpha}\eta^{\alpha(k-1)\beta\dot\alpha(k-2)} + e^\alpha{}_{\dot\beta}\eta^{\alpha(k-2)\dot\alpha(k-1)\dot\beta}, \\
\delta\Omega^{\alpha(k),\dot\alpha(k-2)} &= D\eta^{\alpha(k),\dot\alpha(k-2)} + \lambda^2 e^\alpha{}_{\dot\beta}\xi^{\alpha(k-1)\dot\alpha(k-2)\dot\beta}, \\
\delta\Omega^{\alpha(k-2),\dot\alpha(k)} &= D\eta^{\alpha(k-2),\dot\alpha(k)} + \lambda^2 e_\beta{}^{\dot\alpha}\xi^{\alpha(k-2)\beta\dot\alpha(k-1)}.
\end{aligned}
$$

A remarkable property of frame-like formulation is the possibility to construct gauge invariant objects that generalize the torsion and curvature in gravity:

$$
\begin{aligned}
\mathcal{T}^{\alpha(k-1)\dot\alpha(k-1)} &= Df^{\alpha(k-1)\dot\alpha(k-1)} + e_\beta{}^{\dot\alpha}\Omega^{\alpha(k-1)\beta\dot\alpha(k-2)} + e^\alpha{}_{\dot\beta}\Omega^{\alpha(k-2)\dot\alpha(k-1)\dot\beta}, \\
\mathcal{R}^{\alpha(k),\dot\alpha(k-2)} &= D\Omega^{\alpha(k),\dot\alpha(k-2)} + \lambda^2 e^\alpha{}_{\dot\beta}f^{\alpha(k-1)\dot\alpha(k-2)\dot\beta}, \\
\mathcal{R}^{\alpha(k-2),\dot\alpha(k)} &= D\Omega^{\alpha(k-2),\dot\alpha(k)} + \lambda^2 e_\beta{}^{\dot\alpha}f^{\alpha(k-2)\beta\dot\alpha(k-1)}.
\end{aligned}
$$

To simplify the construction of supermultiplets, we did not introduce any supertransformations for the auxiliary fields $\Omega$. Instead, all calculations were done up to the terms proportional to the auxiliary field equations of motion, which is equivalent to the following "zero torsion conditions":

$$
\mathcal{T}^{a(k-1)} \approx 0 \quad \Rightarrow \quad e_\beta{}^{\dot\alpha}\mathcal{R}^{\alpha(k-1)\beta\dot\alpha(k-2)} + e^\alpha{}_{\dot\beta}\mathcal{R}^{\alpha(k-2)\dot\alpha(k-1)\dot\beta} \approx 0. \tag{2}
$$

As for the supertransformations for the dynamical fields $f$, the corresponding variation of the Lagrangian can be compactly written as follows:

$$
(-1)^k \delta\mathcal{L}_k = -i2\mathcal{R}^{\alpha(k-1)\beta\dot\alpha(k-2)}e_\beta{}^{\dot\beta}\delta f_{\alpha(k-1)\dot\alpha(k-2)\dot\beta} + h.c.
$$

Now let us turn to massless fields with half-integer spin $k + 1/2 \geq 3/2$, which are described by 1-form $\Phi^{\alpha(k)\dot\alpha(k-1)}$, $\Phi^{\alpha(k-1)\dot\alpha(k)}$ [46]. To be Majorana fields, they must satisfy the reality condition:

$$
(\Phi^{\alpha(k)\dot\alpha(k-1)})^\dagger = \Phi^{\alpha(k-1)\dot\alpha(k)}.
$$

The corresponding Lagrangian has the form:

$$
\begin{aligned}
(-1)^k \mathcal{L}_{k+\frac{1}{2}} = {}& \Phi_{\alpha(k-1)\beta\dot\alpha(k-1)}e^\beta{}_{\dot\beta}D\Phi^{\alpha(k-1)\dot\alpha(k-1)\dot\beta} \\
&+ \epsilon_{k+\frac{1}{2}}\frac{\lambda}{2}\big[(k+1)\Phi_{\alpha(k-1)\beta\dot\alpha(k-1)}E^\beta{}_\gamma\Phi^{\alpha(k-1)\gamma\dot\alpha(k-1)} \\
&\qquad - (k-1)\Phi_{\alpha(k)\dot\alpha(k-2)\dot\beta}E^{\dot\beta}{}_{\dot\gamma}\Phi^{\alpha(k)\dot\alpha(k-2)\dot\gamma} + h.c.\big].
\end{aligned} \tag{3}
$$

The Lagrangian is invariant under gauge transformations:

$$
\begin{aligned}
\delta\Phi^{\alpha(k)\dot\alpha(k-1)} &= D\xi^{\alpha(k)\dot\alpha(k-1)} + e_\beta{}^{\dot\alpha}\eta^{\alpha(k)\beta\dot\alpha(k-2)} + \epsilon_{k+\frac{1}{2}}\lambda e^\alpha{}_{\dot\beta}\xi^{\alpha(k-1)\dot\alpha(k-1)\dot\beta}, \\
\delta\Phi^{\alpha(k-1)\dot\alpha(k)} &= D\xi^{\alpha(k-1)\dot\alpha(k)} + e^\alpha{}_{\dot\beta}\eta^{\alpha(k-2)\dot\alpha(k)\dot\beta} + \epsilon_{k+\frac{1}{2}}\lambda e_\beta{}^{\dot\alpha}\xi^{\alpha(k-1)\beta\dot\alpha(k-1)},
\end{aligned}
$$

where $\epsilon_{k+\frac{1}{2}} = \pm 1$. Note that the above consideration does not fix a sign of $\epsilon_{k+\frac{1}{2}}$. As in the integer spin case, we can construct the gauge invariant curvatures:

$$
\begin{aligned}
\mathcal{F}^{\alpha(k)\dot\alpha(k-1)} &= D\Phi^{\alpha(k)\dot\alpha(k-1)} + \epsilon_{k+\frac{1}{2}}\lambda e^\alpha{}_{\dot\beta}\Phi^{\alpha(k-1)\dot\alpha(k-1)\dot\beta}, \\
\mathcal{F}^{\alpha(k-1)\dot\alpha(k)} &= D\Phi^{\alpha(k-1)\dot\alpha(k)} + \epsilon_{k+\frac{1}{2}}\lambda e_\beta{}^{\dot\alpha}\Phi^{\alpha(k-1)\beta\dot\alpha(k-1)}.
\end{aligned}
$$

Then, the Lagrangian variation can be compactly written as follows:

$$(-1)^k \delta \mathcal{L}_{k+\frac{1}{2}} = -\mathcal{F}_{\alpha(k-1)\beta\dot{\alpha}(k-1)} e^{\beta}{}_{\dot{\beta}} \delta \Phi^{\alpha(k-1)\dot{\alpha}(k-1)\dot{\beta}} + h.c.$$

In both the bosonic and in fermionic cases, the variation of the higher spin Lagrangians is completely expressed in geometric terms.

### 3. Minimal $N = 1$ Supermultipets

In this section, we present the minimal massless $N = 1$ supermultiplets in 4D AdS. The off-shell formulation of the N = 1 higher superspin free Lagrangian theory in 4D AdS space was first developed in [56] at the very beginning of superfield formalism, and its component form was derived from superfield theory. In the frame-like approach, massless 4D AdS higher spin supermultiplets were considered in [37], extending the results of [45] in 4D Minkowski space. In the next sections, they play the role of building blocks for constructing extended supermultiplets.

*3.1. Higher Superspins*

The **Supermultiplet** $(k + 1/2, k)$ contains two massless fields with spin $k$ and spin $k + 1/2$. They are described by the fields:

$$f^{\alpha(k-1)\dot{\alpha}(k-1)}, \quad \Omega^{\alpha(k)\dot{\alpha}(k-2)}, \quad \Omega^{\alpha(k-2)\dot{\alpha}(k)},$$

and:

$$\Phi^{\alpha(k)\dot{\alpha}(k-1)}, \quad \Phi^{\alpha(k-1)\dot{\alpha}(k)},$$

respectively. The corresponding supertransformations are written in the form:

$$\delta f^{\alpha(k-1)\dot{\alpha}(k-1)} = a\Phi^{\alpha(k-1)\beta\dot{\alpha}(k-1)}\zeta_{\beta} - \bar{a}\Phi^{\alpha(k-1)\dot{\alpha}(k-1)\dot{\beta}}\zeta_{\dot{\beta}},$$

$$\delta\Phi^{\alpha(k)\dot{\alpha}(k-1)} = b\Omega^{\alpha(k)\dot{\alpha}(k-2)}\zeta^{\dot{\alpha}} + cf^{\alpha(k-1)\dot{\alpha}(k-1)}\zeta^{\alpha},$$

$$\delta\Phi^{\alpha(k-1)\dot{\alpha}(k)} = \bar{b}\Omega^{\alpha(k-2)\dot{\alpha}(k)}\zeta^{\alpha} + \bar{c}f^{\alpha(k-1)\dot{\alpha}(k-1)}\zeta^{\dot{\alpha}},$$

where $a$, $b$, and $c$ are the complex parameters. The parameters of the $N = 1$ supertransformations $\zeta^{\alpha}$, and $\zeta^{\dot{\alpha}}$ satisfy the relation:

$$D\zeta^{\alpha} = -\lambda e^{\alpha}{}_{\dot{\beta}}\zeta^{\dot{\beta}}, \qquad D\zeta^{\dot{\alpha}} = -\lambda e_{\beta}{}^{\dot{\alpha}}\zeta^{\beta}. \tag{4}$$

Note that we do not introduce any supertransformation for the auxiliary field $\Omega$ here, nor further on, since its calculations were done up to equations of motion (2). The invariance of the Lagrangian $\delta(\mathcal{L}_k + \mathcal{L}_{k+\frac{1}{2}}) = 0$ requires restrictions on the coefficients:

$$a = i\frac{(k-1)}{4}\bar{b}, \quad c = \lambda b, \quad b = \epsilon_{k+\frac{1}{2}}\bar{b}, \quad \epsilon_{k+\frac{1}{2}} = \pm 1.$$

The free complex parameter $b$ can be taken as purely real or purely imaginary. In AdS space, it relates the sign of the mass-like term for the fermionic field and the parity of the bosonic field. The two cases $\epsilon_{k+\frac{1}{2}} = +1/-1$ correspond to different $N = 1$ massless supermultiplets with parity-even/odd boson. To fix parameter $b$, one must calculate the commutator of two supertransformations on the bosonic field:

$$\frac{1}{\rho}[\delta_1, \delta_2]f^{\alpha(k-1)\dot{\alpha}(k-1)} = \Omega^{\alpha(k-1)\beta\dot{\alpha}(k-2)}\xi_{\beta}{}^{\dot{\alpha}} + \Omega^{\alpha(k-2)\dot{\alpha}(k-1)\dot{\beta}}\xi^{\alpha}{}_{\dot{\beta}}$$

$$+\lambda\left(f^{\alpha(k-2)\beta\dot{\alpha}(k-1)}\eta^{\alpha}{}_{\beta} + f^{\alpha(k-1)\dot{\alpha}(k-2)\dot{\beta}}\eta^{\dot{\alpha}}{}_{\dot{\beta}}\right), \tag{5}$$

where:

$$\xi_\beta{}^{\dot\alpha} = i(\zeta_1^{\dot\alpha}\zeta_{2\beta} - \zeta_2^{\dot\alpha}\zeta_{1\beta}), \quad \eta^\alpha{}_\beta = i(\zeta_1^\alpha\zeta_{2\beta} - \zeta_2^\alpha\zeta_{1\beta}), \tag{6}$$

and:

$$\rho = \frac{(k-1)}{4}\bar{b}b.$$

We can see that the commutator of these supertransformations is a combination of translation with parameter $\xi^{\alpha\dot\alpha}$ and Lorentz rotation with parameters $\eta^{\alpha\beta}$ and $\eta^{\dot\alpha\dot\beta}$. This means that the two corresponding supercharges $Q_\alpha$ and $Q_{\dot\alpha}$ satisfy the commutation relations of $N = 1$ AdS superalgebra:

$$\begin{aligned}
\{Q_\alpha, Q_{\dot\beta}\} &\sim P_{\alpha\dot\beta}, \\
\{Q_\alpha, Q_\beta\} &\sim \lambda M_{\alpha\beta}, \\
\{Q_{\dot\alpha}, Q_{\dot\beta}\} &\sim \lambda M_{\dot\alpha\dot\beta},
\end{aligned}$$

where $P_{\alpha\dot\alpha}$, $M_{\alpha(2)}$, amd $M_{\dot\alpha(2)}$ are the AdS generators.

The **Supermultiplet** $(k, k-1/2)$ contains massless integer spin $k$ and half-integer spin $k - 1/2$. The corresponding fields are:

$$f^{\alpha(k-1)\dot\alpha(k-1)}, \quad \Omega^{\alpha(k)\dot\alpha(k-2)}, \quad \Omega^{\alpha(k-2)\dot\alpha(k)},$$

and:

$$\Phi^{\alpha(k-1)\dot\alpha(k-2)}, \quad \Phi^{\alpha(k-2)\dot\alpha(k-1)}.$$

Supertransformations under the equations of motion for the auxiliary field $\Omega$ (2) can be written as:

$$\begin{aligned}
\delta f^{\alpha(k-1)\dot\alpha(k-1)} &= a'\Phi^{\alpha(k-1)\dot\alpha(k-2)}\zeta^{\dot\alpha} - \bar{a}'\Phi^{\alpha(k-2)\dot\alpha(k-1)}\zeta^\alpha, \\
\delta\Psi^{\alpha(k-1)\dot\alpha(k-2)} &= b'\Omega^{\alpha(k-1)\beta\dot\alpha(k-2)}\zeta_\beta + c'f^{\alpha(k-1)\dot\alpha(k-2)\dot\beta}\zeta_{\dot\beta}, \\
\delta\Psi^{\alpha(k-2)\dot\alpha(k-1)} &= \bar{b}'\Omega^{\alpha(k-2)\dot\alpha(k-1)\dot\beta}\zeta_{\dot\beta} + \bar{c}'f^{\alpha(k-2)\beta\dot\alpha(k-1)}\zeta_\beta.
\end{aligned}$$

The Lagrangian invariance $\delta(\mathcal{L}_k + \mathcal{L}_{k-\frac{1}{2}}) = 0$ gives:

$$a' = \frac{i}{4(k-1)}\bar{b}', \quad c' = \lambda b', \quad b' = \epsilon_{k-\frac{1}{2}}\bar{b}', \quad \epsilon_{k-\frac{1}{2}} = \pm 1.$$

Again, the free parameter $b'$ can be purely real/imaginary. It corresponds to two different $N = 1$ massless supermultiplets with parity-even/odd boson. Calculating the commutator of two supertransformations, which is equal (5), we fix can $b'$:

$$\rho = \frac{1}{4(k-1)}\bar{b}'b'.$$

### 3.2. Low Superspins

The **Supermultiplet** $(3/2, 1)$ contains a massless field with spin $3/2$, which is described by the 1-forms $\Phi^\alpha$ and $\Phi^{\dot\alpha}$ with the Lagrangian (3) at $k = 1$:

$$\mathcal{L}_{\frac{3}{2}} = -\Psi_\beta e^\beta{}_{\dot\beta}D\Psi^{\dot\beta} - \epsilon_{\frac{3}{2}}\lambda[\Psi_\beta E^\beta{}_\gamma\Psi^\gamma + h.c.].$$

Massless spin 1 is described by dynamical 1-form $f$ and auxiliary 0-forms $W^{\alpha(2)}$ and $W^{\dot\alpha(2)}$. The corresponding Lagrangian looks like:

$$\frac{1}{i}\mathcal{L}_1 = 2EW_{\alpha(2)}W^{\alpha(2)} + E_{\alpha(2)}W^{\alpha(2)}Df + h.c.$$

It is evident that the Lagrangian is invariant under gauge transformations:

$$\delta f = D\xi, \qquad \delta W^{\alpha(2)} = 0.$$

We did not introduce any supertransformation for the auxiliary field $W^{\alpha(2)}$, since the calculations were done up to the equations of motion, which are equivalent to the condition:

$$\mathcal{T} = Df + 2(E_{\alpha(2)}W^{\alpha(2)} + E_{\dot{\alpha}(2)}W^{\dot{\alpha}(2)}) \approx 0. \tag{7}$$

As an consequence of the above condition, we obtained the relation:

$$E_{\alpha(2)}DW^{\alpha(2)} + E_{\dot{\alpha}(2)}DW^{\dot{\alpha}(2)} \approx 0.$$

Then, the supertransformations can be rewritten in the form:

$$\begin{aligned}
\delta f &= a\Phi^{\alpha}\zeta_{\alpha} - \bar{a}\Phi^{\dot{\alpha}}\zeta_{\dot{\alpha}}, \\
\delta\Phi^{\alpha} &= be_{\beta\dot{\beta}}W^{\alpha\beta}\zeta^{\dot{\beta}} + cf\zeta^{\alpha}, \\
\delta\Phi^{\dot{\alpha}} &= \bar{b}e_{\beta\dot{\beta}}W^{\dot{\alpha}\dot{\beta}}\zeta^{\beta} + \bar{c}f\zeta^{\dot{\alpha}}.
\end{aligned}$$

The condition of the Lagrangian invariance $\delta(\mathcal{L}_1 + \mathcal{L}_{\frac{3}{2}}) = 0$ under equation $\mathcal{T} \approx 0$ yields:

$$a = -i\frac{\bar{b}}{2}, \quad c = -\frac{\lambda}{2}b, \quad b = \epsilon_{\frac{3}{2}}\bar{b}, \quad \epsilon_{\frac{3}{2}} = \pm 1.$$

The commutator of the two supertransformations has the form:

$$\frac{1}{\rho}[\delta_1, \delta_2]f = -2e_{\beta\dot{\beta}}(W^{\alpha\beta}\zeta_{\alpha}{}^{\dot{\beta}} + W^{\dot{\alpha}\dot{\beta}}\zeta^{\beta}{}_{\dot{\alpha}}), \tag{8}$$

where $\zeta^{\alpha\dot{\alpha}}$ is the same as in (6) and $\rho = \frac{\bar{b}b}{4}$.

The **Supermultiplet** $(1, 1/2)$ contains the massless spin 1 described in the same way as in the previous case and the massless spin 1/2 described by 0-forms $Y^{\alpha}$ and $Y^{\dot{\alpha}}$. The Lagrangian for spin 1/2 field has the form:

$$\mathcal{L} = -Y_{\alpha}E^{\alpha}{}_{\dot{\alpha}}DY^{\dot{\alpha}}.$$

Note that unlike the higher spin fermionic fields, there is no mass-like term in the above Lagrangian. The supertransformations, up to the equations of motion for the auxiliary field $W^{\alpha\beta}$, are written as follows:

$$\begin{aligned}
\delta f &= a'e_{\alpha\dot{\alpha}}Y^{\alpha}\zeta^{\dot{\alpha}} - \bar{a}'e_{\alpha\dot{\alpha}}Y^{\dot{\alpha}}\zeta^{\alpha}, \\
\delta Y^{\alpha} &= b'W^{\alpha\beta}\zeta_{\beta}, \\
\delta Y^{\dot{\alpha}} &= \bar{b}'W^{\dot{\alpha}\dot{\beta}}\zeta_{\dot{\beta}}.
\end{aligned}$$

The Lagrangian invariance $\delta(\mathcal{L}_1 + \mathcal{L}_{\frac{1}{2}}) = 0$ under equation $\mathcal{T} \approx 0$ (7) yields:

$$a' = -\frac{i}{4}\bar{b}'.$$

Calculating the commutator of the superetransformations leads to the relation (8) and allows to fix the parameter $\rho = \frac{\bar{b}'b'}{8}$.

The **Supermultiplet** $(1/2, 0)$ contains the massless spin $1/2$ and one massless spin 0, which is described by the dynamical 0-form $W$ and the auxiliary 0-form $W^{\alpha\dot\alpha}$. The Lagrangian for spin 0 has the form:

$$\frac{1}{i}\mathcal{L} = -\frac{1}{2}EW_{\alpha\dot\alpha}W^{\alpha\dot\alpha} - E_{\alpha\dot\alpha}W^{\alpha\dot\alpha}DW + 2\lambda^2EW^2.$$

Using the equation of motion for the auxiliary field $W^{\alpha\dot\alpha}$:

$$\mathcal{W} = DW + e_{\alpha\dot\alpha}W^{\alpha\dot\alpha} \approx 0,$$

one gets the relation:

$$e_{\alpha\dot\alpha}DW^{\alpha\dot\alpha} \approx 0 \quad \Rightarrow \quad E^{\alpha}{}_{\dot\gamma}DW^{\beta\dot\gamma} \approx \frac{1}{2}\varepsilon^{\alpha\beta}E_{\gamma\dot\gamma}DW^{\gamma\dot\gamma}.$$

We used the following anzac for the supertransformations:

$$\begin{aligned}
\delta W &= a_0 Y^{\alpha}\zeta_{\alpha} - \bar{a}_0 Y^{\dot\alpha}\zeta_{\dot\alpha}, \\
\delta Y^{\alpha} &= b_0 W^{\alpha\dot\alpha}\zeta_{\dot\alpha} + c_0 W\zeta^{\alpha}, \\
\delta Y^{\dot\alpha} &= \bar{b}_0 W^{\alpha\dot\alpha}\zeta_{\alpha} + \bar{c}_0 W\zeta^{\dot\alpha},
\end{aligned}$$

with the set of arbitrary complex parameters $a_0$, $b_0$, and $c_0$. The invariance of the Lagrangian $\delta(\mathcal{L}_0 + \mathcal{L}_{\frac{1}{2}}) = 0$ under the equation $\mathcal{W} \approx 0$ places restrictions on the parameters:

$$a_0 = \frac{i}{2}\bar{b}_0, \quad c_0 = \lambda b_0.$$

The commutator of two supertransformations for the spin 0 field has the form:

$$\frac{1}{\rho}[\delta_1, \delta_2]W = W^{\alpha\dot\alpha}\xi_{\alpha\dot\alpha},$$

where $\xi_{\alpha\dot\alpha}$ is the parameter of translation (6) and $\rho = \bar{b}_0 b_0$. The parameter $\beta$ can be taken as purely real/imaginary depending on whether the supermultiplet has a parity-even/odd spin 0 field.

The **chiral supermultiplet** $(1/2, 0_+, 0_-)$ contains one massless spin $1/2$ and two massless parity-even/odd spins $0_+/0_-$. In this case, the spin 0 is described by the complex scalar field $W$. The corresponding supertransformations have the form:

$$\delta W = 2a_0 Y^{\alpha}\zeta_{\alpha} \qquad\qquad \delta Y^{\alpha} = b_0 W^{\alpha\dot\alpha}\zeta_{\dot\alpha} + c_0 W\zeta^{\alpha}, \qquad (9)$$

$$\delta\bar{W} = -2\bar{a}_0 Y^{\dot\alpha}\zeta_{\dot\alpha} \qquad\qquad \delta Y^{\dot\alpha} = \bar{b}_0 \bar{W}^{\alpha\dot\alpha}\zeta_{\alpha} + \bar{c}_0 \bar{W}\zeta^{\dot\alpha}, \qquad (10)$$

where:

$$a_0 = \frac{i}{2}\bar{b}_0, \quad c_0 = \lambda b_0.$$

The commutators of the supertransformations are written as follows:

$$\frac{1}{\rho}[\delta_1, \delta_2]W = W^{\alpha\dot\alpha}\xi_{\alpha\dot\alpha}, \qquad \frac{1}{\rho}[\delta_1, \delta_2]\bar{W} = \bar{W}^{\alpha\dot\alpha}\xi_{\alpha\dot\alpha},$$

where $\rho = \bar{b}_0 b_0$. To re-denote the complex field in the form:

$$W = W_+ + iW_- \qquad W^{\alpha\dot\alpha} = W_+^{\alpha\dot\alpha} + iW_-^{\alpha\dot\alpha},$$

$$\bar{W} = W_+ - iW_- \qquad \bar{W}^{\alpha\dot\alpha} = W_+^{\alpha\dot\alpha} - iW_-^{\alpha\dot\alpha},$$

then, at real $b_0$, the field $W_+$ is a parity-even spin 0 field, while $W_-$ is a parity-odd one. At imaginary $b_0$, the $W_+$ is a parity-odd field and $W_-$ is a parity-even one.

## 4. *N*-Extended Supermultiplets

In this section, we consider the massless *N*-extended higher spin supermultiplets in 4D AdS. As we pointed out in the Introduction, for the given maximal integer or half-integer spin $k$ in the supermultiplet, the parameter *N* satisfies the relation $N \leq 4k$. For each spin $k$, we described the field contents and the corresponding field variables. Then, we derived the supertransformations and showed that the specially defined free Lagrangians are invariant under these transformations. Finally, we proved that the constructed supertransformations form the on-shell closed *N*-extended 4D AdS superalgebras.

*4.1. $N \leq 2k - 3$*

In this case, the massless supermultiplets contain the massless fields with spins:

$$ k, k - \frac{1}{2}, k - 1, ..., k - \frac{N-1}{2}, k - \frac{N}{2}, $$

where $k$ is an arbitrary integer or half-integer. We can write it compactly as:

$$ k - \frac{m}{2}, \quad m = 0, 1, ..., N. $$

The number of massless fields with the given spin $k - \frac{m}{2}$ is equal to $\frac{N!}{m!(N-m)!}$. One can see that the minimal spin equals $\frac{3}{2}$ in the boundary case $N = 2k - 3$. Thus, all massless fields entering extended supermultiplets are uniformly described in Section 1.

Let us introduce the bosonic field variables:

$$ f_{k-\frac{m}{2},i[m]}{}^{\alpha(k-\frac{m+2}{2})\dot{\alpha}(k-\frac{m+2}{2})}, \quad \Omega_{k-\frac{m}{2},i[m]}{}^{\alpha(k-\frac{m}{2})\dot{\alpha}(k-\frac{m+4}{2})}, $$

and the fermionic ones:

$$ \Phi_{k-\frac{m}{2},i[m]}{}^{\alpha(k-\frac{m+1}{2})\dot{\alpha}(k-\frac{m+3}{2})}. $$

where the first lower index denotes the spin of the field, and the compact index $i[m] = [i_1 i_2 ... i_m]$ denotes the antisymmetric combination of indices $i = 1, 2, ...N$ and corresponds to the antisymmetric representation of the internal symmetry group $SO(N)$. If the maximal spin $k$ is an integer, then $m$ takes the even values $0, 2, ..., 2[\frac{N}{2}]$ for the bosonic fields and the odd values $1, 3, ..., 2[\frac{N-1}{2}] + 1$ for the fermionic ones. In the case of the maximal half-integer spin $k$, the parameter $m$ takes even values for fermions and odd one for bosons.

The generic anzatz for the linear supertransformations was chosen in the following form with a set of arbitrary complex coefficients $a_m$, $a'_m$, $b_m$, $b'_m$, $c_m$, and $c'_m$:

$$
\begin{aligned}
\delta f_{k-\frac{m}{2},i[m]}{}^{\alpha(k-\frac{m+2}{2})\dot\alpha(k-\frac{m+2}{2})} = {}& a'_m \Phi_{k-\frac{m+1}{2},i[m]j}{}^{\alpha(k-\frac{m+2}{2})\dot\alpha(k-\frac{m+4}{2})}\zeta^{j\dot\alpha} \\
& -\bar a'_m \Phi_{k-\frac{m+1}{2},i[m]j}{}^{\alpha(k-\frac{m+4}{2})\dot\alpha(k-\frac{m+2}{2})}\zeta^{j\alpha} \\
& +a_m \Phi_{k-\frac{m-1}{2},i[m-1]}{}^{\alpha(k-\frac{m+2}{2})\beta\dot\alpha(k-\frac{m+2}{2})}\zeta_{i\beta} \\
& -\bar a_m \Phi_{k-\frac{m-1}{2},i[m-1]}{}^{\alpha(k-\frac{m+2}{2})\dot\alpha(k-\frac{m+2}{2})\dot\beta}\zeta_{i\dot\beta},
\end{aligned}
\tag{11}
$$

$$
\begin{aligned}
\delta \Phi_{k-\frac{m}{2},i[m]}{}^{\alpha(k-\frac{m+1}{2})\dot\alpha(k-\frac{m+3}{2})} = {}& b_m \Omega_{k-\frac{m+1}{2},i[m]j}{}^{\alpha(k-\frac{m+1}{2})\dot\alpha(k-\frac{m+5}{2})}\zeta^{j,\dot\alpha} \\
& +b'_m \Omega_{k-\frac{m-1}{2},i[m-1]}{}^{\alpha(k-\frac{m+1}{2})\beta\dot\alpha(k-\frac{m+3}{2})}\zeta_{i,\beta} \\
& +c_m f_{k-\frac{m+1}{2},i[m]j}{}^{\alpha(k-\frac{m+3}{2})\dot\alpha(k-\frac{m+3}{2})}\zeta^{j,\alpha} \\
& +c'_m f_{k-\frac{m-1}{2},i[m-1]}{}^{\alpha(k-\frac{m+1}{2})\dot\alpha(k-\frac{m+3}{2})\dot\beta}\zeta_{i,\dot\beta}.
\end{aligned}
\tag{12}
$$

where $\zeta_i^\alpha$ and $\zeta_i^{\dot\alpha}$ are the parameters of the extended supertransformations satisfying the conditions (4). The Lagrangian is defined as $\mathcal{L} = \sum_m \mathcal{L}_{k-\frac{m}{2}}$, where $\mathcal{L}_{k-\frac{m}{2}}$ is the Lagrangian for the free field with spin $k-\frac{m}{2}$. Invariance of the Lagrangian under these supertransformations leads to restrictions on the arbitrary parameters:

$$
a_m = \frac{i(k-\frac{m+2}{2})}{4}\bar b_{m-1} \quad c_m = \lambda b_m, \quad b_m = \epsilon_{k-\frac{m}{2}}\bar b_m,
$$

$$
a'_m = \frac{i}{4(k-\frac{m+2}{2})}\bar b'_{m+1} \quad c'_m = \lambda b'_m, \quad b'_m = \epsilon_{k-\frac{m}{2}}\bar b'_m.
$$

In these relations, $\epsilon_{k-\frac{m}{2}} = +1$ or $\epsilon_{k-\frac{m}{2}} = -1$ for any $m$ depending on the parity of the corresponding field. This means that there are two families of parameters, $b_m$ and $b'_m$. In order to relate them to one another, the supertransformation algebras (11) and (12) need to be closer. This yields the condition:

$$
a_m b_{m-1} - a'_m b'_{m+1} = 0.
\tag{13}
$$

The calculation of the commutator for the two supertransformations (11) allows to obtain the result:

$$
\begin{aligned}
\frac{1}{\rho}[\delta_1,\delta_2]f_{k-\frac{m}{2},i[m]}{}^{\alpha(k-\frac{m+2}{2})\dot\alpha(k-\frac{m+2}{2})} = {}& \Omega_{k-\frac{m}{2},i[m]}{}^{\alpha(k-\frac{m+2}{2})\beta\dot\alpha(k-\frac{m+4}{2})}\xi_\beta{}^{\dot\alpha} \\
& +\Omega_{k-\frac{m}{2},i[m]}{}^{\alpha(k-\frac{m+4}{2})\dot\alpha(k-\frac{m+2}{2})\dot\beta}\xi^\alpha{}_{\dot\beta} \\
& +\lambda\big(f_{k-\frac{m}{2},i[m]}{}^{\alpha(k-\frac{m+2}{2})\dot\alpha(k-\frac{m+4}{2})\dot\beta}\eta_{\dot\beta}{}^{\dot\alpha} \\
& +f_{k-\frac{m}{2},i[m]}{}^{\alpha(k-\frac{m+4}{2})\beta\dot\alpha(k-\frac{m+2}{2})}\eta_\beta{}^\alpha\big) \\
& +\lambda f_{k-\frac{m}{2},i[m-1]j}{}^{\alpha(k-\frac{m+2}{2})\dot\alpha(k-\frac{m+2}{2})}z^j{}_i,
\end{aligned}
\tag{14}
$$

where:

$$
\xi_\beta{}^{\dot\alpha} = i(\zeta_{1j,\beta}\zeta_2^{j\,\dot\alpha} - \zeta_{2j,\beta}\zeta_1^{j\,\dot\alpha}), \quad \eta_{\dot\beta}{}^{\dot\alpha} = i(\zeta_{1j,\dot\beta}\zeta_2^{j\,\dot\alpha} - \zeta_{2j,\dot\beta}\zeta_1^{j\,\dot\alpha})
\tag{15}
$$

$$
z^j{}_i = i(\zeta_1^{j,\beta}\zeta_{2i\beta} - \zeta_2^{j,\beta}\zeta_{1i\beta} + \zeta_1^{j,\dot\beta}\zeta_{2i\dot\beta} - \zeta_2^{j,\dot\beta}\zeta_{1i\dot\beta}), \quad z^{ij} = -z^{ji}.
\tag{16}
$$

One can see that commutator (14) is equal to combinations of translations, Lorentz rotations, and internal $SO(N)$ transformations with parameters $\xi^{\alpha\dot\alpha}$, $\eta^{\alpha\beta}$, and $z^{ij}$, respectively. We calculated

such a commutator only for the bosonic fields. For the fermionic fields, the results were the same; however, the computations became more complicated and tedious. From (13) and (14), we obtained restrictions for the parameters:

$$\bar{b}_{m-1}b_{m-1} = \frac{4\rho}{(k - \frac{m+2}{2})}, \quad \bar{b}'_{m+1}b'_{m+1} = 4\rho(k - \frac{m+2}{2}).$$

The form of the above commutator proves that the supercharges $Q_\alpha{}^i$ and $Q_{\dot\alpha}{}^i$ corresponding to the supertransformations (11) and (12) satisfy the commutation relations of the extended AdS superalgebra:

$$\begin{aligned}
\{Q_\alpha^i, Q_{\dot\beta}^j\} &\sim \delta^{ij} P_{\alpha\dot\beta}, \\
\{Q_\alpha^i, Q_\beta^j\} &\sim \lambda(\delta^{ij} M_{\alpha\beta} + \frac{1}{2}\varepsilon_{\alpha\beta} T^{ij}), \\
\{Q_{\dot\alpha}^i, Q_{\dot\beta}^j\} &\sim \lambda(\delta^{ij} M_{\dot\alpha\dot\beta} + \frac{1}{2}\varepsilon_{\dot\alpha\dot\beta} T^{ij}),
\end{aligned} \tag{17}$$

where $P_{\alpha\dot\alpha}$, $M_{\alpha(2)}$, and $M_{\dot\alpha(2)}$ are the AdS generators and $T^{ij} = -T^{ji}$ refers to the generators of the internal $SO(N)$ group symmetry.

*4.2. $2k - 3 < N \leq 2k$*

In order to go beyond $N > 2k - 3$, we should include massless fields with lower spins to the supermultiplets. In the case of $N = 2k - 2$, it is sufficient to add the massless spin 1 and the corresponding fields:

$$f_{1,i[2k-2]}, \quad W_{i[2k-2]}{}^{\alpha(2)}.$$

Analogically, in the cases of $N = 2k - 1$ and $N = 2k$, we also should add the massless fields with spin $\frac{1}{2}$:

$$Y_{i[2k-1]}{}^\alpha, \quad Y_{i[2k-1]}{}^{\dot\alpha},$$

and set of complex fields for spin 0:

$$W_{i[2k]}, \quad W_{i[2k]}{}^{\alpha\dot\alpha} \qquad \bar{W}_{i[2k]}, \quad \bar{W}_{i[2k]}{}^{\alpha\dot\alpha}.$$

In this case, the anzatz for supertransformations with a set of arbitrary parameters looks like:

$$\begin{aligned}
\delta\Phi_{\frac{3}{2},i[2k-3]}{}^\alpha &= b'_{2k-3}\Omega_{2,i[2k-4]}{}^{\alpha\beta}\zeta_{i,\beta} + c'_{2k-3}f_{2,i[2k-4]}{}^{\alpha\dot\beta}\zeta_{i,\dot\beta} \\
&\quad + b_{2k-3}e_{\beta\dot\beta}W_{i[2k-3]j}{}^{\alpha\beta}\zeta^{j\dot\beta} + c_{2k-3}f_{1,i[2k-3]j}\zeta^{j\alpha},
\end{aligned} \tag{18}$$

$$\delta f_{1,i[2k-2]} = a_{2k-2}\Phi_{\frac{3}{2},i[2k-3]}{}^\alpha \zeta_{i\alpha} + a'_{2k-2}e_{\alpha\dot\alpha}Y_{i[2k-2]j}{}^\alpha \zeta^{j\dot\alpha}, \tag{19}$$

$$\begin{aligned}
\delta Y_{i[2k-1]}{}^\alpha &= b'_{2k-1}W_{i[2k-2]}{}^{\alpha\beta}\zeta_{i\beta} \\
&\quad + b_{2k-1}W_{i[2k-1]j}{}^{\alpha\dot\alpha}\zeta^j{}_{\dot\alpha} + c_{2k-1}W_{i[2k-1]j}\zeta^{j\alpha},
\end{aligned} \tag{20}$$

$$\delta W_{i[2k]} = 2a_{2k}Y_{i[2k-1]}{}^\alpha\zeta_{i\alpha}, \quad \delta\bar{W}_{i[2k]} = -2\bar{a}_{2k}Y_{i[2k-1]}{}^{\dot\alpha}\zeta_{i\dot\alpha}. \tag{21}$$

The Lagrangian is defined as a sum of the Lagrangians for all of the fields in the supermultiplet. Invariance of the Lagrangian yields restrictions for the arbitrary parameters:

$$a_{2k-2} = -\frac{i\bar{b}_{2k-3}}{2}, \quad c_{2k-3} = -\frac{\lambda}{2}b_{2k-3}, \quad b_{2k-3} = \epsilon_{\frac{3}{2}}\bar{b}_{2k-3}, \quad \epsilon_{\frac{3}{2}} = \pm 1,$$

$$a'_{2k-2} = -\frac{i}{4}\bar{b}'_{2k-1}, \quad a_{2k} = \frac{i}{2}\bar{b}_{2k-1}, \quad c_{2k-1} = \lambda b_{2k-1}.$$

As a result, there are three arbitrary complex parameters, namely, $b_{2k-3}$, $b'_{2k-1}$, and $b_{2k-1}$, which can be either purely real or purely imaginary depending on the parity of the bosonic fields entering the supermultiplets being even or odd. We fixed them by the requirement that commutators for spins 1 and 0 are closed. This gives the condition:

$$a_{2k-2}b_{2k-3} - a'_{2k-2}b'_{2k-1} = 0. \tag{22}$$

Then, the commutators for the spin 1 field has the following form up to the gauge transformations:

$$\frac{1}{\rho}[\delta_1, \delta_2] f_{i[2k-2]} = -2e_{\alpha\dot\alpha}(W_{i[2k-2]}{}^{\alpha\beta}\xi_\beta{}^{\dot\alpha} + W_{i[2k-2]}{}^{\dot\alpha\dot\beta}\xi^\alpha{}_{\dot\beta}) + \lambda f_{i[2k-3]j}z^j{}_i, \tag{23}$$

$$\frac{1}{\rho}[\delta_1, \delta_2] W_{i[2k]} = W_{i[2k]}{}^{\alpha\dot\alpha}\xi_{\alpha\dot\alpha}, \tag{24}$$

where $\xi^{\alpha\dot\alpha}$ and $z^{ij}$ are the parameters of the translations and internal $SO(N)$ symmetry defined by (15) and (16). The relations (22)–(24) lead to the following conditions for the parameters:

$$\bar{b}_{2k-3}b_{2k-3} = 4\rho, \quad \bar{b}'_{2k-1}b'_{2k-1} = 8\rho, \quad \bar{b}_{2k-1}b_{2k-1} = \rho.$$

Using the commutators (23) and (24), we can show that the corresponding supercharges satisfy the relations (17).

*4.3. $2k < N < 4k$*

To extend the supersymmetry further, i.e., to consider the case of $N > 2k$, we Appendix A. Notations and Conventions

$$\frac{1}{2}, 1, \frac{m}{2} - k \quad m = 2k + 3, 2k + 4, ..., N - 1, N.$$

Therefore, we introduced the additional field variables for spin $\frac{1}{2}$:

$$Y_{i[2k+1]}{}^\alpha, \quad Y_{i[2k+1]}{}^{\dot\alpha}.$$

For spin 1:

$$f_{1,i[2k+2]}, \quad W_{i[2k+2]}{}^{\alpha(2)}.$$

For higher spins:

$$f_{\frac{m}{2}-k,i[m]}{}^{\alpha(\frac{m-2}{2}-k)\dot\alpha(\frac{m-2}{2}-k)}, \quad \Omega_{\frac{m}{2}-k,i[m]}{}^{\alpha(\frac{m}{2}-k)\dot\alpha(\frac{m-4}{2}-k)}, \quad m = 2k + 4, ..., 2[\frac{N}{2}]$$

$$\Phi_{\frac{m}{2}-k,i[m]}{}^{\alpha(\frac{m-1}{2}-k)\dot\alpha(\frac{m-3}{2}-k)}, \quad m = 2k + 3, ..., 2[\frac{N-1}{2}] + 1.$$

An additional anzatz for the supertransformations was chosen in the following form with a set of arbitrary parameters:

$$\delta f_{\frac{m}{2}-k,i[m]}{}^{\alpha(\frac{m-2}{2}-k)\dot{\alpha}(\frac{m-2}{2}-k)} = a'_m \Phi_{\frac{m+1}{2}-k,i[m]j}{}^{\alpha(\frac{m-2}{2}-k)\beta\dot{\alpha}(\frac{m-2}{2}-k)}\zeta^j{}_\beta$$

$$+ a_m \Phi_{\frac{m-1}{2}-k,i[m-1]}{}^{\alpha(\frac{m-2}{2}-k)\dot{\alpha}(\frac{m-4}{2}-k)}\zeta_i{}^{\dot{\alpha}} + h.c. \quad m \geq 2k+4,$$

$$\delta \Phi_{\frac{m}{2}-k,i[m]}{}^{\alpha(\frac{m-1}{2}-k)\dot{\alpha}(\frac{m-3}{2}-k)} = b_m \Omega_{\frac{m+1}{2}-k,i[m]j}{}^{\alpha(\frac{m-1}{2}-k)\beta\dot{\alpha}(\frac{m-3}{2}-k)}\zeta^j{}_\beta$$

$$+ b'_m \Omega_{\frac{m-1}{2}-k,i[m-1]}{}^{\alpha(\frac{m-1}{2}-k)\dot{\alpha}(\frac{m-5}{2}-k)}\zeta_i{}^{\dot{\alpha}}$$

$$+ c_m f_{\frac{m+1}{2}-k,i[m]j}{}^{\alpha(\frac{m-1}{2}-k)\dot{\alpha}(\frac{m-3}{2}-k)\dot{\beta}}\zeta^j{}_{\dot{\beta}}$$

$$+ c'_m f_{\frac{m-1}{2}-k,i[m]}{}^{\alpha(\frac{m-3}{2}-k)\dot{\alpha}(\frac{m-3}{2}-k)}\zeta_i{}^{\alpha} \quad m \geq 2k+5,$$

$$\delta \Phi_{\frac{3}{2},i[2k+3]}{}^{\alpha} = b_{2k+3}\Omega_{2,i[2k+3]j}{}^{\alpha\beta}\zeta^j{}_\beta + c_{2k+3}f_{2,i[2k+3]j}{}^{\alpha\dot{\beta}}\zeta^j{}_{\dot{\beta}}$$

$$+ b'_{2k+3}e_{\beta\dot{\beta}}W_{i[2k+2]}{}^{\alpha\beta}\zeta_i{}^{\dot{\beta}} + c'_{2k+3}f_{1,i[2k+2]}\zeta_i{}^{\alpha},$$

$$\delta f_{1,i[2k+2]} = a'_{2k+2}\Phi_{\frac{3}{2},i[2k+2]j}{}^{\alpha}\zeta^j{}_\alpha + a_{2k+2}e_{\alpha\dot{\alpha}}Y_{i[2k+1]}{}^{\alpha}\zeta_i{}^{\dot{\alpha}},$$

$$\delta Y_{i[2k+1]}{}^{\alpha} = b_{2k+1}W_{i[2k+1]j}{}^{\alpha\beta}\zeta^j{}_\beta$$

$$+ b'_{2k+1}\bar{W}_{i[2k]}{}^{\alpha\dot{\alpha}}\zeta_{i\dot{\alpha}} + c'_{2k+1}\bar{W}_{i[2k]}\zeta_i{}^{\alpha},$$

$$\delta W_{i[2k]} = -2\bar{a}'_{2k}Y_{i[2k]j}{}^{\dot{\alpha}}\zeta^j{}_{\dot{\alpha}} \quad \delta \bar{W}_{i[2k]} = 2a'_{2k}Y_{i[2k]j}{}^{\alpha}\zeta^j{}_\alpha.$$

The Lagrangian is defined as a sum of the Lagrangians for all of the fields in the supermultiplet. The condition of invariance of the Lagrangian invariance yields restrictions on the arbitrary parameters:

$$a_m = \frac{i}{4(\frac{m-2}{2}-k)}\bar{b}_{m-1}, \quad c_m = \lambda b_m, \quad b_m = \epsilon_{\frac{m}{2}-k}\bar{b}_m,$$

$$a'_m = \frac{i(\frac{m-2}{2}-k)}{4}\bar{b}'_{m+1}, \quad c'_m = \lambda b'_m, \quad b'_m = \epsilon_{\frac{m}{2}-k}\bar{b}'_m,$$

$$a'_{2k+2} = -i\frac{\bar{b}'_{2k+3}}{2}, \quad c'_{2k+3} = -\frac{\lambda}{2}b'_{2k+3}, \quad b'_{2k+3} = \epsilon_{\frac{3}{2}}\bar{b}'_{2k+3}, \quad \epsilon_{\frac{3}{2}} = \pm 1,$$

$$a_{2k+2} = -\frac{i}{4}\bar{b}_{2k+1}, \quad a'_{2k} = \frac{i}{2}\bar{b}'_{2k+1}, \quad c'_{2k+1} = \lambda b'_{2k+1}.$$

To close algebra of the supertransformations, we imposed the conditions:

$$a_m b_{m-1} - a'_m b'_{m+1} = 0, \quad a_{2k+2}b_{2k+1} - a'_{2k+2}b'_{2k+3} = 0, \quad 2a_{2k}b_{2k-1} + 2\bar{a}'_{2k}\bar{b}'_{2k+1} = 0. \tag{25}$$

As a result, the commutators of the supertransformations had the form:

$$
\begin{aligned}
\frac{1}{\rho}[\delta_1, \delta_2] f_{\frac{m}{2}-k, i[m]}{}^{\alpha(k-\frac{m+2}{2})\dot\alpha(k-\frac{m+2}{2})} &= \Omega_{\frac{m}{2}-k, i[m]}{}^{\alpha(k-\frac{m+2}{2})\beta\dot\alpha(k-\frac{m+4}{2})}\xi_\beta{}^{\dot\alpha} \\
&+ \Omega_{\frac{m}{2}-k, i[m]}{}^{\alpha(k-\frac{m+4}{2})\dot\alpha(k-\frac{m+2}{2})\dot\beta}\xi^\alpha{}_{\dot\beta} \\
&+ \lambda \big( f_{\frac{m}{2}-k, i[m]}{}^{\alpha(k-\frac{m+2}{2})\dot\alpha(k-\frac{m+2}{2})\dot\beta}\eta_{\dot\beta}{}^{\dot\alpha} \\
&+ f_{\frac{m}{2}-k, i[m]}{}^{\alpha(k-\frac{m+4}{2})\beta\dot\alpha(k-\frac{m+2}{2})}\eta_\beta{}^\alpha \big) \\
&+ \lambda f_{\frac{m}{2}-k, i[m-1]j}{}^{\alpha(k-\frac{m+2}{2})\dot\alpha(k-\frac{m+2}{2})}z^j{}_i,
\end{aligned}
\tag{26}
$$

$$
\frac{1}{\rho}[\delta_1, \delta_2] f_{1, i[2k+2]} = -2e_{\alpha\dot\alpha}\big( W_{i[2k+2]}{}^{\alpha\beta}\xi_\beta{}^{\dot\alpha} + W_{i[2k+2]}{}^{\dot\alpha\dot\beta}\xi^\alpha{}_{\dot\beta}\big) + \lambda f_{i[2k-3]j}z^j{}_i,
$$

$$
\frac{1}{\rho}[\delta_1, \delta_2] W_{i[2k]} = W_{i[2k]}{}^{\alpha\dot\alpha}\xi_{\alpha\dot\alpha} + \lambda W_{i[2k-1]j}z^j{}_i,
$$

where $\xi^{\alpha\dot\alpha}$, $\eta^{\alpha\beta}$, and $z^{ij}$ are defined by (15) and (16). The form of the above commutators and relation (25) allowed to find additional restrictions for the arbitrary parameters:

$$
\bar{b}_{m-1}b_{m-1} = 4\rho\Big(\frac{m-2}{2} - k\Big), \quad \bar{b}'_{m+1}b'_{m+1} = \frac{4\rho}{(\frac{m-2}{2} - k)},
$$

$$
\bar{b}'_{2k+3}b'_{2k+3} = 4\rho, \quad \bar{b}_{2k+1}b_{2k+1} = 8\rho, \quad \bar{b}'_{2k+1}b'_{2k+1} = \rho.
$$

The commutator (26) allowed to calculate the algebra of the supercharges with the form (17). As a result, we obtained the on-shell *N*-extended component free Lagrangian formulations for supermultiplets with $2k < N < 4k$.

### 4.4. N = 4k

Now, we consider a special case of maximal *N*-extended supersymmetry with the highest spin $k$ in a supermultiplet. Such a supermultiplet contains the massless fields with all spins from $k$ to 0. The field variables are the same as in the $N = 2k$ case, but now, $i = 1, 2 \cdots, 4k$:

$$
f_{k-\frac{m}{2}, i[m]}{}^{\alpha(k-\frac{m+2}{2})\dot\alpha(k-\frac{m+2}{2})}, \quad \Omega_{k-\frac{m}{2}, i[m]}{}^{\alpha(k-\frac{m}{2})\dot\alpha(k-\frac{m+4}{2})},
$$

for the bosonic higher spin fields and:

$$
\Phi_{k-\frac{m}{2}, i[m]}{}^{\alpha(k-\frac{m+1}{2})\dot\alpha(k-\frac{m+3}{2})},
$$

for the fermionic higher spin fields. For spin 1, we introduced the fields:

$$
f_{1, i[2k-2]}, \quad W_{i[2k-2]}{}^{\alpha(2)}.
$$

For spin 1/2, we introduced the fields:

$$
Y_{i[2k-1]}{}^\alpha, \quad Y_{i[2k-1]}{}^{\dot\alpha}.
$$

Moreover, we introduced a set of complex fields for spin 0:

$$
W_{i[2k]}, \quad W_{i[2k]}{}^{\alpha\dot\alpha} \quad \bar{W}_{i[2k]}, \quad \bar{W}_{i[2k]}{}^{\alpha\dot\alpha}
$$

which are subject to the condition:

$$
W_{i[2k]} = \frac{1}{(2k)!}\mathcal{E}_{i[2k]}{}^{j[2k]}\bar{W}_{j[2k]},
\tag{27}
$$

where $k$ is an arbitrary integer. As before, all field variables are totally antisymmetric over the indices $i = 1, 2, ...N$. Totally antisymmetric invariant tensors $\mathcal{E}_{i[4k]} = \mathcal{E}_{[i_1 i_2 ... i_{4k}]}$ are normalized as $\mathcal{E}_{12...4k} = 1$.

The supertransformations in the case under consideration are the same as in the cases of $N = 2k$, (11), (12), and (18)–(20). Meanwhile, for the spin 0 components, the supertransformations compatible with (27) look like:

$$
\begin{aligned}
\delta W_{i[2k]} &= 2a_{2k} Y_{i[2k-1]}{}^{\alpha} \zeta_{i\alpha} - \frac{2\bar{a}_{2k}}{(2k-1)!} \mathcal{E}_{i[2k]}{}^{j[2k]} Y_{j[2k-1]}{}^{\dot{\alpha}} \zeta_{j\dot{\alpha}}, \\
\delta \bar{W}_{i[2k]} &= -2\bar{a}_{2k} Y_{i[2k-1]}{}^{\dot{\alpha}} \zeta_{i\dot{\alpha}} + \frac{2a_{2k}}{(2k-1)!} \mathcal{E}_{i[2k]}{}^{j[2k]} Y_{j[2k-1]}{}^{\alpha} \zeta_{j\alpha}.
\end{aligned}
\tag{28}
$$

In this case, again, the algebra of supercharges has the form (17) and the Lagrangian is invariant under the transformations (11), (12), (18)–(20), and (28).

To conclude this subsection, one notes that we studied here only the case of maximal integer spin $k$. In the case of maximal half-integer spin, the above consideration is not applicable, since the relation (27) is inconsistent with the half-integer $k$. This case requires a special analysis. The matter is that the consistent usage of the generic scheme described Section 4.3 leads to double of all of the fields in the supermultiplet. In the case of maximal integer spin, this doubling can be avoided if one imposes, in particular, the condition (27). In the case of maximal half-integer spin, such a condition is impossible We are grateful to Yu.M. Zinoviev for discussion of this question.

## 5. Summary and Prospects

In this paper, we studied the field realization of arbitrary $N$-extended massless supermultiplets in 4D AdS space. For the arbitrary highest integer or half-integer spin $k$ fields entering the supermultiplets, we realized the on-shell supersymmetric component Lagrangian formulations under the condition $N < 4k$ and defined the higher superspin field Lagrangians as the sums of the free Lagrangians for all of the fields in the supermultiplets. We constructed the supertransformations that form the closed on-shell algebras and leave invariant the Lagrangians. It was shown that the commutators of two such supertransformations form the $N$-extended AdS superalgebra, i.e., they are combinations of the translations, Lorentz rotations, and internal $SO(N)$-transformations. Moreover, we realized maximally extended supermultiplets, where $N = 4k$ in the case of the highest integer spin $k$, and were constructed the corresponding supertransformations.

We hope that our results can be helpful for the construction of extended massive higher spin and massless infinite spin supermultiplets and their Lagrangian formulations, extending the results of the $N = 1$ case [37–39]. Moreover, we pointed out some other open problems in the free supersymmetric higher spin field theory. First of all, this is a problem of the superfield Lagrangian formulation of the 4D $N = 1$ supersymmetric massive higher superspin fields. The corresponding massless theories were constructed in [53,54,56]. As regards massive theories, there are only partial examples of the higher superspin massive $N = 1$ superfield models [60–65]. The problem of formulating extended supersymmetric higher spin theories in terms of unconstrained superfields is completely open. At present, the only case where such a possibility can, in principle, be realized is 4D $N = 2$ supersymmetry, where the harmonic superfield approach [66] allows to construct field models in terms of $N = 2$ unconstrained superfields. We hope to study some of these open problems in forthcoming papers.

**Author Contributions:** The contributions of all of the authors were the same, all of whom worked together to develop the present manuscript. All authors read and agreed to the published version of the manuscript.

**Acknowledgments:** The authors thank S.M. Kuzenko for useful comments and Yu.M. Zinoviev for fruitful discussions. The work was partially supported by the Ministry of Science and Higher Education of the Russian Federation, project No. FEWF-2080-0003, and an RFBR grant, project No. 18-02-00153. T.V.S. acknowledges partial support from the President of Russia grant for young scientists, No. MK-1649.2019.2.

**Conflicts of Interest:** The authors declare no conflict of interest.

## Appendix A. Notations and Conventions

In the paper, we adopted "condensed notation" for the indices. Namely, if some expression contains $n$ consecutive indices, denoted by the same letter with different numbers (e.g., $\alpha_1, \alpha_2, \ldots \alpha_n$) and is symmetric on them, we simply wrote the letter, with the number $n$ in parentheses if $n > 1$ (e.g., $\alpha(n)$). For example:

$$\Phi^{\alpha_1,\alpha_2,\alpha_3} = \Phi^{\alpha(3)}, \qquad \zeta^{\alpha_1}\Omega^{\alpha_2\alpha_3} = \zeta^\alpha \Omega^{\alpha(2)}. \tag{A1}$$

We defined symmetrization over indices as the sum of the minimal number of terms necessary without a normalization multiplier.

We used the multispinor formalism in four dimensions (see, e.g., [67]). Every vector index was transformed into a pair of spinor indices: $V^\mu \sim V^{\alpha\dot\alpha}$, where $\alpha, \dot\alpha = 1, 2$. Dotted and undotted indices were transformed into one another under the hermitian conjugation:

$$\left(\Omega^{\alpha\dot\alpha(2)}\right)^\dagger = \Omega^{\alpha(2)\dot\alpha}. \tag{A2}$$

The spin-tensors, i.e., fields with odd number of indices, were Grassmannian. For example:

$$A^{\alpha(2)\dot\alpha}\eta^\alpha = -\eta^\alpha A^{\alpha(2)\dot\alpha}. \tag{A3}$$

Under the hermitian conjugation, the order of fields was reversed:

$$\left(A^{\alpha(2)\dot\alpha}\eta^\alpha\right)^\dagger = \eta^\alpha A^{\alpha(2)\dot\alpha} = -A^{\alpha(2)\dot\alpha}\eta^\alpha. \tag{A4}$$

The metrics for the spinor indices comprised an antisymmetric bispinor $\epsilon_{\alpha\beta}$ and an inverse one $\epsilon^{\alpha\beta}$:

$$\epsilon_{\alpha\beta}\zeta^\beta = -\zeta_\alpha, \qquad \epsilon^{\alpha\beta}\zeta_\beta = \zeta^\alpha, \tag{A5}$$

similarly for dotted indices.

In frame-like formalism, two bases, namely, the world one and the local one, are used. We denotef the local basis vectors as $e^{\alpha\dot\alpha}$, while the world indices were omitted; all of the fields were assumed to be of differential forms. Similarly, all of the products of the forms were exterior with respect to the world indices. In this paper, we used the basis forms, i.e., antisymmetrized products of basis vectors $e^{\alpha\dot\alpha}$. The forms were the 2-form $E^{\alpha(2)} + h.c.$, the 3-form $E^{\alpha\dot\alpha}$, and the 4-form $E$. The transformation law of the forms under the hermitian conjugation is:

$$(e^{\alpha\dot\alpha})^\dagger = e^{\alpha\dot\alpha}, \qquad (E^{\alpha(2)})^\dagger = E^{\dot\alpha(2)}, \qquad (E^{\alpha\dot\alpha})^\dagger = -E^{\alpha\dot\alpha}, \qquad (E)^\dagger = -E. \tag{A6}$$

The covariant AdS derivative satisfied the following normalization conditions:

$$D \wedge D\Omega^{\alpha(m)\dot\alpha(n)} = -2\lambda^2(E^\alpha{}_\beta \Omega^{\alpha(m-1)\beta\dot\alpha(n)} + E^{\dot\alpha}{}_{\dot\beta}\Omega^{\alpha(m)\dot\alpha(n-1)\dot\beta}). \tag{A7}$$

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
