# Peer review of "Lagrangian Formulation of Free Arbitrary N-Extended Massless Higher Spin Supermultiplets in 4D AdS Space"

_symmetry, doi:10.3390/sym12122052_

Round 1

Reviewer 1 Report

The authors consider massless higher spin fields in 4-dimensional anti-de Sitter space
and construct N-extended supermultiplets in the component formalism. They also show that the Lagrangian, defined as the sum of the Lagrangians of the single higher spin fields, is on-shell invariant under N-extended supersymmetry.

The paper is clearly written and is of timely interest. There are some really minor issues that the authors should adjust:
1) Supposedly, Sections 2 and 3 are reviews of higher spin fields in 4-dimensional anti-de Sitter space and of the N=1 supermultiplets, respectively. Citations to the original work should be given at the appropriate places in the text, not only in the introduction.

2) In Section 3, the supermultiplets are specified by a series of integer and half-integers numbers. Sometimes they are in decreasing order, sometimes increasing. This notation should be consistent.

3) The notation of the parameters $\alpha$ in the transformations in section 3.1 clashes with the use of $\alpha$ as multi-index.

4) "anzac" should be "ansatz", which derives from the german "Ansatz", meaning starting point.

Author Response

Response to Reviewer 1 Comments.

We thank the Reviewer for the comments and suggestions.

Point 1: Supposedly, Sections 2 and 3 are reviews of higher spin fields in 4-dimensional anti-de Sitter space and of the N=1 supermultiplets, respectively. Citations to the original work should be given at the appropriate places in the text, not only in the introduction.

Response 1: In the beginning of the Sections 2 and 3 we added citations to the original work. In the beginning of the Section 3 we also added additional comments on the first realization of N=1 massless higher spin supermultiplets in 4D, AdS space.

Point 2: In Section 3, the supermultiplets are specified by a series of integer and half-integers numbers. Sometimes they are in decreasing order, sometimes increasing. This notation should be consistent.

Response 2: In Section 3, we uniformly specified the supermultiplets by pair of integer and half-integer numbers in decreasing order.

Point 3:  The notation of the parameters $\alpha$ in the transformations in section 3.1 clashes with the use of $\alpha$ as multi-index.

Response 3: We substituted the notation of the parameters $\alpha$ in the supertransformations in section 3.1 for the notation $a$ to avoid the clashes with the use of $\alpha$ as multi-index. The same was done for the parameters $\beta$ and $\gamma$ in the supertransformations that was substituted for $b$ and $c$ respectively. We did this substitution everywhere in the text, not only in section 3.1.  

Point 4: "anzac" should be "ansatz", which derives from the german "Ansatz", meaning starting point.  

Response 4: It was corrected everywhere in the text.

Reviewer 2 Report

The main result is to construct actions for free supermultiplets of massless higher-spin fields, to write down and check the supersymmetry transformations and the closure of the on-shell superalgebra. The paper is clearly written and delivers interesting enough and explicit results that can be useful for the study of higher-spin theories. I recommend the paper for publication in its present form.

Author Response

We thank the reviewer for considering our work and approving it for publication.